# Portland Cement-Based Grouts Enhanced with Basalt Fibers for Post-Tensioned Concrete Duct Filling

**DOI:** 10.3390/ma16072842

**Published:** 2023-04-03

**Authors:** José R. Zapata-Padilla, César A. Juárez-Alvarado, Alejandro Durán-Herrera, Miguel A. Baltazar-Zamora, Bernardo. T. Terán-Torres, Francisco R. Vázquez-Leal, José M. Mendoza-Rangel

**Affiliations:** 1Facultad de Ingeniería Civil, Universidad Autónoma de Nuevo León, Av. Universidad S/N, San Nicolás de los Garza 66455, Mexico; 2Facultad de Ingeniería Civil-Xalapa, Universidad Veracruzana, Lomas del Estadio S/N, Zona Universitaria Xalapa, Veracruz 91000, Mexico

**Keywords:** basalt microfibers, grout, prestressed concrete, permeability, Portland cement

## Abstract

In post-tensioned systems, grouts act as a last line of defense to prevent the penetration of harmful compounds such as chlorides, moisture and other substances that cause corrosion in the prestressing steel. For this reason, improving grouts results in the enhancement of the overall durability of the structure. In this study, the physical properties of grouts with basalt microfiber additions in the amounts of 0.03, 0.07 and 0.10% with respect to the mix volume were evaluated. The fresh properties included flowability and unit mass. Specimens were fabricated to evaluate drying shrinkage, compressive strength, air permeability and rapid permeability to chloride ions. The incorporation of basalt microfibers showed a beneficial effect on the physical properties of the grout by increasing the drying shrinkage resistance and decreasing the permeability compared to the reference mix and two commercial dry prepackaged grouts. The optimal grout mix was the one with a percentage of basalt microfibers of 0.10%, which decreased drying shrinkage by 15.98% at 14 days compared to the reference mix, and permeability to chloride ions decreased by 10.82% compared to the control mix.

## 1. Introduction

A prestressed concrete element can be defined as a structural concrete component in which steel elements permanently induce internal compressive stresses to counteract, to some extent, the tensile stresses caused by the applied loads [1,2,3,4]. In post-tensioned concrete (PTC), the prestressing steel is installed in ducts and tensioned once the concrete has hardened and reached a specific strength [3,5,6]. The elements can be constructed as prefabricated units or can be cast in place. In post-tensioned and bonded construction, the tendon is generally in contact with a Portland cement grout. This grout is injected into a polyethylene or galvanized steel conduit embedded in the concrete [7]. This system provides corrosion protection in post-tensioned construction due to additional barriers [8,9,10], because in addition to the concrete, the conduit and grout prevent the entry of various harmful ions and compounds such as chlorides, moisture and other substances that initiate corrosion by reaching the steel strands. However, this superior protection is only achieved if the duct is completely filled with grout. Several problems have been attributed to the lack of or inadequate grout injection [11,12,13], such as the formation of voids in the ducts. This situation is of great concern because these air cavities make the steel tendons highly vulnerable to corrosion [8].

The cementitious grout bonds to the prestressing steel and surrounding concrete and provides corrosion protection to the steel, as it is a barrier against moisture and chloride penetration (physical protection) and produces an alkaline environment for the tendon (chemical protection) [8]. For the cementitious grout to satisfy these requirements for physical protection, it must be designed for low permeability, since permeability is a basic indicator of durability, by providing the pathway for permeation of aggressive agents [14]. However, it has been reported that cementitious materials are brittle, porous and crack-prone, as they have low tensile strength, low toughness and small ultimate tensile strain, and therefore are susceptible to cracking [15,16]. Porosities are reported to appear from the moment the cement paste reacts with water; at this stage, products are formed which replace the original cement particles and expand in the space originally occupied by water. The microstructure thus contains unreacted cement particles, hydration products and pores. The pore structure is even more complex because the main product, calcium silicate hydrate (C-S-H), is itself porous. Thus, two types of porosity have been defined: (1) capillary porosity, which is the largest porosity remaining between the hydrating cement particles, and (2) gel porosity, which is the finest porosity, approximately all of which is within the C-S-H. There is no strict distinction based on the size between these types, but generally speaking, gel pores are less than 10 nm in diameter. Fine porosity is involved in shrinkage and creep mechanisms, while larger pores influence properties such as strength and permeability [17]. Porosity and pore size distribution are the most direct parameters for the permeability of cement pastes [18]. Permeability is directly related to the continuity of the fluid passage (pore diameters of 120 or 160 nm), and porosity is a measure of the ratio of pores to the total volume of pastes [16]. If a cement-based specimen has high porosity and the pores are connected, the material has a higher permeability. On the contrary, if the pores are not connected, the permeability of the materials tends to decrease [16,19]. Thus, it can be observed that pore connectivity has a considerable relationship between permeability and durability [14,15,16,20].

It has been reported that cracks in cement pastes generally interconnect flow paths and increase the permeability of the composite. Moreover, this increase in the permeability of cement pastes due to the progression of cracks allows more water or aggressive chemical ions to penetrate the concrete, facilitating its deterioration [21]. In the service stage, concretes are in different stress states due to different external loads, which causes the distribution and propagation of microcracks, and this significantly alters the porous structure [14]. However, cement pastes are prone to cracking from early ages, either due to volumetric shrinkage during the first few hours after placement (plastic shrinkage), or after a longer period of time, after the concrete has hardened (drying shrinkage) [22]. Therefore, the ability to control microcracking is crucial to improve the durability of concrete. Currently, research on concrete crack control mainly involves the addition of fibers to concrete, which is an effective method to reduce the brittleness of concrete, delay the development of cracks and improve the hardness [15,22,23,24,25,26,27,28,29,30,31,32,33]. The addition of randomly distributed short fibers to concrete has been reported to be an effective method to mitigate plastic shrinkage cracking [22]. Fibers are effective in this regard for two reasons: first, they reduce overall shrinkage deformations and reduce the possibility of tensile stresses exceeding tensile strength, and second, fibers can restrict their development if they occur [22,34]. The addition of any fiber with a diameter less than 40 μm and an aspect ratio greater than 200, in volume fractions of 0.2% to 0.4%, should effectively eliminate plastic shrinkage cracking in concrete [22,35,36,37,38,39,40,41]. Therefore, a wide variety of fibers have now been shown to be beneficial in this regard, including steel, glass, basalt, various synthetic fibers (polypropylene, polyethylene, polyvinyl and carbon) and various natural fibers (sisal, coir, flax and cellulose) [15,22,23,24,25,26,27,28,29,30,32,33,40,42,43,44,45,46].

As expected, Portland cement grouts are prone to cracking, but cracking can be reduced by incorporating fibers into the mix [47], since they provide better control of matrix cracking [47,48,49,50]. Some authors report that basalt microfiber (BF) in its current state of development can improve concrete durability by preventing early microcracks due to plastic shrinkage, and this effect is attributed to the reduction of the magnitude of free shrinkage, since basalt microfibers have a powerful bonding effect with the matrix and effectively restrict the development of micro- and macrocracks with their bridging effect [22,38,51,52]. Moreover, it has been reported that the risk of cracking is more prominent when the w/c ratio is reduced in the mix [22]. Other studies report that basalt fibers without any protective coating lack long-term durability in the alkaline concrete environment [53,54]. Until this problem is overcome, one effective use of BFs is to prevent microcracking in the plastic stage of concrete and at early ages. Basalt microfiber is a fiber composed of natural basalt, which is environmentally friendly, non-toxic and uses cheap and abundant domestic raw materials, which can save production and energy costs [11]. Due to its excellent performance, this fiber has attracted great attention in multiple industries [48,55]. Compared with polyethylene fiber-reinforced concrete, basalt fiber-reinforced concrete is not only suitable for cost saving [55], but also has better performance in cement bond strength because its chemical property is more similar to cement [55,56,57]. These characteristics of fiber make it attractive for creating sustainable grouts and more durable post-tensioned structures by improving its performance in aggressive environments and increasing its lifespan. Basalt microfiber has also been reported to decrease the flowability of cement mortars [11,48], and it is expected to have the same effect on grouts; therefore, an amount of superplasticizer additive should be added to the mix to meet the minimum flowability needed for a grout to ensure complete filling of ducts and satisfactorily coating the prestressing steel [8].

In this study, basalt fiber (BF) was selected to improve grouts in post-tensioned systems due to the reported benefits of this fiber in mortars and concretes. In addition, other authors have reported the improvement of grouts with other types of fiber such as polypropylene, but no information has been found on grouts specifically reinforced with BF, so this was considered an area of opportunity to investigate. In the design of these grouts, low water–cement (w/c) ratios were selected to establish a dense matrix. Since the water contents were low, the flowability was also low; hence, a superplasticizing admixture (SPA) was used to obtain the proper flowability for grouts. Basalt microfibers and a shrinkage-reducing admixture (SRA) were also added to evaluate their effects on volumetric changes or drying shrinkage, given that this is an important parameter at early ages to avoid microcracking that can significantly alter permeability. As expected, the flowability was found to decrease with the addition of BF, and the maximum amount of BF to meet the minimum required flowability for a grout was found to be 0.10% of the volume. Mixes were made with different amounts of FB to find an optimum content; these were a control mix with no BF, and additions of 0.03, 0.07 and 0.10% by volume. These mixtures were also compared with two prepackaged commercial dry grouts and mixed according to the manufacturer’s instructions. Based on the results, it was found that the basalt microfiber decreases drying shrinkage more than an SRA does. It was also found that the laboratory-designed grouts have lower permeability than a grout without BF reinforcement, and that in turn these mixtures have lower permeability than the commercial prepackaged grouts.

## 2. Materials and Methods

### 2.1. Materials

#### 2.1.1. Cement (OPC)

The cement used was provided in bulk by a local cement plant. Moreover, it was of the Ordinary Portland Cement (OPC) type, which corresponds to cement with at least 95% clinker + gypsum and a maximum of 5% minority [58]. It was also a high initial strength cement, Type III according to ASTM C150 [59]. The standard compressive strength was 30 MPa at 3 days and 40 MPa at 28 days. Figure 1 shows the distribution and particle size, and as expected, approximately 95% of the cement particles (D95) are smaller than 55 μm, and the average particle size (D50) is 15.45 μm. The chemical composition and physical properties of OPC are described in Table 1. Figure 2a shows the CPO used.

#### 2.1.2. Commercial Grout 1 (CG-1)

CG-1 is a mortar used in general construction processes, with fluid and semi-fluid consistencies, composed of cement, aggregates with controlled granulometry, and fluidizing and water-reducing additives adequately dosed to control volume changes. Its chemical composition and dry density are shown in Table 1. The CG-1 is distributed in 22.0 kg bags and the manufacturer recommends 3.50 kg of water for a fluid consistency and to produce 0.012 m^3^ of the mix.

#### 2.1.3. Commercial Grout 2 (CG-2)

CG-2 is a grout designed for applications where high strength performance, positive expansion, non-staining characteristics and extreme flowability are required. It contains only natural aggregates and an expansive cementitious binder. Its chemical composition and dry density are presented in Table 1. CG-2 is distributed in 22.7 kg bags and the manufacturer recommends 4.50 kg of water for a fluid consistency to produce 0.013 m³ of the mix.

#### 2.1.4. Fine Aggregate (FA)

A regionally available fine aggregate, of the crushed limestone type, was used. This aggregate was used without any post-processing. To determine the granulometry of the fine aggregate and its fineness modulus, the ASTM C136 [60] test was carried out. The water absorption and relative density were determined using the test ASTM C128 [61]. The results of the granulometry are shown in Figure 3, and the physical properties are presented in Table 2. Figure 2a shows the FA used.

#### 2.1.5. Superplasticizer Additive (SPA)

A high-range water-reducing, hyperfluidizing, polycarboxylate-based additive was used. It complies with ASTM C494 [62] type A and F as a high-range water reducer. It does not contain added chloride ions that can promote corrosion within concrete. It is an amber-colored liquid with a density of 1.11 g/cm^3^ and a solids percentage of 81.73%. The manufacturer recommends a dosage of this admixture in the range of 4 to 15 cm^3^/kg of cement. Preliminary tests were carried out to analyze the influence of the w/c ratio on the flowability of the cement paste, and the effectiveness of SPA as a water reducer at the recommended minimum content of 4 cm^3^/kg. Figure 4 shows the results of two series of pastes. Paste-1 was made with cement and variable amounts of water, while Paste-2-SPA was made with cement, variable amounts of water and a fixed amount of ASP, at a dose of 4 cm^3^/kg. The flowability was measured in the pan according to the ASTM C1437 [63] procedure. It was found that in both pastes, the flowability had a logarithmic trend as a function of the w/c ratio. The water reduction rate of the SPA was variable according to the w/c ratio. Considering a target flowability of 130%, a w/c ratio of 0.40 was required for Paste-1 and a w/c ratio of 0.23 for Paste-2-SPA, i.e., the reduction rate was 0.17 L of water per kg of cement. Figure 2b shows the SPA used.

#### 2.1.6. Shrinkage-Reducing Additive (SRA)

This additive acts directly on the mechanisms that cause shrinkage at the time of cement hydration, reducing the capillary tension of water in the concrete pore. This action substantially reduces drying shrinkage. It does not contain added chlorides and does not favor the corrosion of steel. It is a colorless liquid with a density of 0.97 g/cm^3^ and a percentage of solids of 79.70%. To reduce shrinkage, the manufacturer recommends a dose of 1% to 2% by weight of the cement. Figure 2b shows the SRA used.

#### 2.1.7. Basalt Microfiber (BF)

Basalt microfibers with approximately constant lengths and diameters were used. The mechanical and physical properties reported by the manufacturer are presented in Table 3. In Figure 2c, the basalt microfibers are shown.

### 2.2. Mixture Proportions

A total of six mixes were produced in this study, including four laboratory-designed mixes and two based on the aforementioned commercial grouts. The experimental grouts included a control mix and three grout mixes with basalt fibers in percentages relative to the volumes of 0.03%, 0.07% and 0.10%. In all the mixes, the water/cement ratio (w/c) was 0.40 and the fine aggregate/cement ratio was 2.43. The SPA and SRA were added in percentages with respect to the weight of the cement. The two commercial grouts were prepared according to the specifications of the manufacturers, and the water/grout ratio was 0.1591 and 0.1982 for CG-1 and CG-2, respectively. Details of the mix proportions are presented in Table 4.

### 2.3. Properties in the Fresh State

The grouts were designed in accordance with the requirements specified in ASTM C1107 [64] and the Post-Tensioning Institute (PTI) grout committee specifications [65]. The fluidity was taken as a starting point for the design. According to the ASTM C939 [66], the recommended fluidity in grouts is 125 to 140%, so 130% was selected as the minimum fluidity. The maximum water/cement ratio (w/c) recommended by the PTI is 0.45 for all types of exposure, in this study a w/c of 0.40 was used and a superplasticizer additive was used to achieve the minimum target fluidity.

#### 2.3.1. Flowability of Grouts

The fluidity was determined on the flow table (Figure 5a) using the ASTM C1437 test method [63] and with the modification for grouts listed in the ASTM C1107 [64]. For grouts in liquid consistency, the flow cone (Figure 5b) of ASTM C939 [66] was used.

#### 2.3.2. Unit Mass of the Mixture and Air Content

Grout mixtures were evaluated according to ASTM C185 test [67], using a container of known mass and volume (Figure 5c) to determine their unit mass and air content, this latter obtained by the difference between the absolute volumetric weight (without air) and measured volumetric weight (with air).

### 2.4. Air Permeability and Densification Level

For this test, cylindrical specimens of 100 mm diameter and 50 mm thickness were used after 28 days of wet curing. For the determination of air permeability, a Torrent permeability meter was used [68]. This equipment is suitable for both laboratory and in situ measurements in a non-destructive manner. The equipment has a double cylindrical chamber cell of approximately 10 cm in diameter, connected to a high vacuum pump, with its corresponding sensors and valves controlled by a central processing unit (Figure 6). The cell is bound to the specimen surface by the vacuum produced by the pump and the sensors measure the loss of that vacuum due to the internal porosity of the grout. The double chamber prevents spurious air currents and allows for the calculation of the permeability coefficient K of the material cover, which is displayed on the screen of the processing unit.

### 2.5. Compressive Strength and Drying Shrinkage

#### 2.5.1. Compressive Strength

Cubic specimens of 50 mm per side were used according to the ASTM C109 [69] test method and with the modifications for grouts described in the ASTM C942 [70]. The specimens were tested at ages 7, 14, 28 and 90 days. The compression test is shown in Figure 7a, and the results are the average of the three specimens.

#### 2.5.2. Drying Shrinkage

Drying shrinkage was measured with 25 mm × 25 mm × 285 mm specimens (Figure 7b), by following the ASTM C596 procedure [71]. The specimens were cured for 3 days and, immediately after, the initial reading was recorded, and the specimens were subsequently placed in the drying room. Drying shrinkage readings were obtained at ages 7, 14, 21, 28, 56 and 70 days. The results are the average of the three specimens. Each result was expressed in terms of the percentage of the calibration length (250 mm).

### 2.6. Durability Index

#### Rapid Chloride Ion Permeability

This test method was performed to determine the electrical conductivity of grouts to provide a rapid indication of their resistance to chloride ion penetration. The test was performed following the ASTM C1202 test procedures [72], and by using cylinders of 100 mm diameter and specimens of 50 mm thickness with 28 days of wet curing. Before the test, the specimens were coated with epoxy paint on their surface contour. The test procedure is shown in Figure 8. In this test, the total charge through the section, in coulomb, shows a relationship towards the resistance of the specimen to chloride ion penetration.

## 3. Results and Discussion

### 3.1. Properties at the Fresh State

#### 3.1.1. Flowability of Grouts

As expected, an increasing amount of basalt fiber decreased the flowability of the grouts; this coincides with what has been reported in previous studies [27,49,73,74,75]. This can be attributed to several factors; for example, some authors mention that it is due to the extra consumption of mixing water and cement paste to cover the surface of the fibers [27,73], others that the network structure of BF prevents the cement paste from segregating and flowing, which increases the viscosity [49,74], and others that agglomeration (balling) of fibers can occur during mixing [75]. Hence, the amount of basalt fiber in the grouts was controlled to achieve a flowable consistency of 125% to 145% on the flow table, with five drops in 3 s as specified by the ASTM C1107 [64]. The mix with the highest amount of fiber obtained the lowest flowability, which corresponds to the mix with 0.10% basalt fibers (BF-0.10), which resulted in a 130.5% flow rate. Commercial grout number one (i.e., CG-1) obtained a plastic consistency and did not qualify for the ASTM C1107 test [64]; thus, it was decided to use the procedure for mortars of the ASTM C1437 test [63] with 25 drops in 15 s, and the flow rate it obtained was 127%. Commercial grout number two (i.e., CG-2) presented a liquid consistency, and consequently, the flow cone procedure of the ASTM C939 test [66] was used, with a flow time of 30 s. The results are summarized in Table 5. It is worth noting that, because the flow table has a diameter of 254 mm (10 in), it was not possible to obtain the flowability value of the mixtures with fiber additions of less than 0.10% FB, as these were more fluid and reached a larger diameter than that measurable in the test.

#### 3.1.2. Unit Mass of the Mixtures and Air Content

The unit mass and the air content of each mixture are shown in Table 5. The results show that the apparent density of the grouts with BF has a decreasing trend with the increase in BF volume content, and in contrast, a proportional increase in void volume is shown. Although the density of BF is slightly higher than that of grout, the density of the mixture decreases with increasing BF content. Guo et al. [74] reported in their study of basalt fibers in concrete mixtures that this phenomenon is due to multiple reasons. The first reason is that BF can cause a loose matrix around the fiber and decrease the compactness of the mix; and the second is the formed network structure of BF prevents the cement paste from segregating and flowing, which causes voids and bubbles in the mix to be more difficult to remove with vibration. All these phenomena increase the amount of air voids in the mix, and the amount of air voids is an influential factor in determining the apparent density of concrete [74,76]. It has also been reported that the addition of BF often introduces defects in the matrix of the material, creating a loose matrix and voids around the fiber [26,74,77], which can explain the decrease in apparent density.

The densities of commercial grouts are shown to be lower than laboratory grouts; this may be explained by the lower density of their materials, their proportions and the higher air content. It may also be due to the types of components of these prepackaged grouts such as the density of aggregates, or their content of supplementary cementitious materials, because the density of a mixture that includes mineral additives is lower compared to one without mineral additives with the same fiber content [74]. Moreover, its powdered chemical additives, such as SPA, can create additional air voids since they contain the components of the air-entraining agent [74].

### 3.2. Air Permeability and Densification Level

The air permeability is determined by the KT constant, and according to the specifications of the equipment manufacturer, the results can be classified by quality index, as shown in Table 6. The results of the air permeability tests are presented in Table 7. All laboratory grouts show a “very good” quality because the air permeability constant (KT) is less than 0.01 (10^−16^ m^2^). In the BF mixes, an improvement over the control mix can be noted because the KT constant is reduced from 0.002 to 0.001 (10^−16^ m^2^). Since the KT value of 0.001 is the minimum reading of the device, with this test it was not possible to observe the influence of the amount of BF on the permeability; however, it was possible to differentiate the permeability between the control mix, the mix with BF and the commercial grouts. In the commercial grouts, it can be observed that CG-1 has a “bad” quality since its KT is higher than 1.0 (10^−16^ m^2^) and that CG-2 showed a “normal” quality, with a KT higher than 0.1 (10^−16^ m^2^). These results show that the BF reduces the interconnectivity of the pores due to microfissures, resulting in a lower air permeability. These results are consistent with the rapid chloride ion permeability test.

### 3.3. Compressive Strength and Drying Shrinkage

#### 3.3.1. Compressive Strength

In post-tensioned construction, the incorporation of grout provides adhesion between the prestressing steel and the concrete, thus considerably increasing the flexural strength, the ductility of the member and the ultimate strength (through better crack distribution) [2,3,5,7]. This bond has minimal effect on the behavior of the member under normal loading conditions, but in the case of overloading it would affect both the nature of cracking and the safety factor against failure of the section [78]. Wang et al. [79] investigated the effects of the spacing between adjacent tendons and the compressive strength of the grout on the maximum bond stress. Figure 9 shows their results on the correlation between adhesion and maximum compression efforts in the grouts. They found that, for three-strand and seven-strand tendons, increasing the grout compressive strength from 25 MPa to 65 MPa resulted in 77.21% and 77.25% increases in the maximum bond stress, respectively. Due to limitations, bond tests were not performed in this study, but based on the compression tests and the report by Wang et al. [79], an estimate of the maximum bond stress of the manufactured grouts can be made.

Dias and Thaumaturgo [27] reported that, for Portland cement concrete (high workability, w/c ratio of 0.80), the addition of basalt microfiber (length of 45 mm and diameter of 9 µm) at a volume fraction of 1.0% and 0.5% caused a −26.4% and −3.9% reduction, respectively, in the 28-day compressive strength. This is because the higher the fiber percentage, the greater the likelihood of creating voids in the matrix or the fibers absorbing an excess of water, resulting in the cement around these fibers having insufficient water to hydrate. Jiang et al. [49] reported that the addition of basalt microfiber (12 mm in length and diameter 20 µm) in a volume fraction of 0.05%, 0.1%, 0.3% and 0.5% in a concrete mix (Slump 172 at 65 mm) can improve the compressive strength by 3.74% to 6.49% at the age of 7 days, relative to the control blend without fiber. However, the increase in compressive strength at an older age is less than that of an earlier age. At 28 days, the change in compressive strength ranged from −0.18% to 4.68%, and the reduction in compressive strength was even more evident at 90 days. The reason may be that the aging of the interface between the fibers and the concrete matrix leads to a drop in the bonding capacity of the substrate, which is because BFs lack long-term durability in the alkaline environment of concrete [53,54]. For this reason, the current applications of BF are to improve the durability of concrete by preventing early cracking due to plastic shrinkage [22].

According to the literature, the percentages of BF in proportions of 0.5% or higher lead to a negative impact on the compressive strength; for proportions lower than 0.5%, there is a slight improvement trend at early ages, but this trend decreases at advanced ages. The addition of the fiber is also reported to reduce material flow.

In this study, through preliminary tests, it was found that to comply with the minimum fluidity of a grout, a maximum BF content of 0.10% of the volume can be used; therefore, it was expected that the compressive strength would increase slightly in samples with BF at early ages and that this improvement would diminish over time. The results of the compression test of the grout mixtures are presented in Figure 10. It is observed that the incorporation of BF does not significantly modify the magnitude of the compressive strength of the grout. The tendency to increase the compressive strength of the grout with the Incorporation of basalt fiber can be found within the 7- to 28-day curing periods. With an additional 90-day cure increase, BF produces no improvement in the compressive strength of the grout. The test results are consistent with the results of previous studies on concrete with basalt fibers [49] and mortars with basalt fibers [50,80]. Nevertheless, it can also be observed that the compressive strengths of the laboratory grouts are higher than both commercial grouts. The correlation between compressive stress and maximum adherence reported by Wang et al. [79], and shown in Figure 9, has an approximately linear trend. It is observed that for grouts with compressive strengths of 25 MPa and 65 MPa and a 3-strand tendon, a bond strength of 3.5 MPa and 6.3 MPa is obtained, respectively. In this study, the grouts obtained compressive strengths that exceed 70 MPa (Figure 10), so it can be estimated by linear extrapolation that the bond stress is approximately 6.65 MPa for a 3-strand tendon and 4.75 MPa for a 7-strand tendon.

#### 3.3.2. Drying Shrinkage

The drying shrinkage curves are presented in Figure 11, and it can be seen that the drying shrinkage increases with curing time for all samples, increasing faster in the early stages and decreasing in the later stages, as expected and as reported by Jiang et al. [50]. The graph shows that the three mixtures containing BF and shrinkage-reducing additive have lower shrinkages than the control mixture (which contained shrinkage-reducing additive but not contain BF). Thus, by comparing the three grouts with BF and the control, it can be estimated that the addition of BF restricts shrinkage more than the shrinkage-reducing admixture by itself. It can also be seen that the three mixtures with BF are close to each other and almost the same; hence, it can be estimated that the three additions of BF lead to similar results. Comparing laboratory grouts with commercial grouts, it is clearly observed that the dry shrinkage of commercial grouts is much higher, which could cause microcracks in these mixes of concern.

One of the main challenges for a post-tensioned concrete duct backfill grout is to avoid cracking that occurs in the hardening process. These cracks occur due to internal stresses from contractions during hardening, in addition to the fact that in post-tensioned ducts, no external water can be provided during grout hardening. The hydration of cement consumes the pore water, resulting in self-drying of the paste and uniform volume shrinkage. These volume contractions become even larger when using w/c ratios lower than 0.42 [76], and in this study, w/c = 0.40 was used. For this reason, it is of great importance to evaluate the volume changes in the grouts studied. According to the results obtained, the addition of basalt fibers to the mix yielded favorable results to counteract these problems. Fiber-reinforced grouts have a matrix composed of cement paste and fiber reinforcement, which spreads the stress through the cracks created in the matrix.

### 3.4. Durability Index

#### Rapid Chloride Ion Permeability

The results of the rapid chloride permeability test (RCPT) of the grout mixtures at the age of 28 days are shown in Figure 12. The control mixture obtained a permeability of 4433.08 Coulombs which corresponds to a high permeability according to the ASTM C1202 classification [72]. As mentioned above, the permeability of the paste is particularly important as the paste covers all the components in the mixture. Permeability is affected by the water–cement ratio (w/c), degree of cement hydration and wet curing period [76]. It is also known that the use of supplementary cementitious materials, such as blast furnace slag, fly ash and silica fume, improves the pore structure and reduces the permeability of hardened concrete [81]. In this investigation, grout mixtures were made with ordinary Portland cement (OPC) without the addition of supplementary cementitious materials, and a w/c ratio of 0.40. A concrete of w/c = 0.40 has a chloride permeability of 5000 Coulombs [82,83], and the mortars have a chloride permeability of 4125 Coulombs [84]. Thus, the chloride permeability result of the control grout is within the expected range.

As shown in Figure 12, the addition of BF in the range of 0.03% to 0.10% by volume resulted in a proportional decrease in chloride ion penetrability. According to the qualitative indications of chloride ion penetrability, based on the measured values of electrical charge passed by the ASTM C1202 method [72], all grouts except BF-3 were classified as “High” permeability. The BF-10 grout was classified with “Moderate” permeability. In this BF-0.10 grout, the electric charge decreased by 10.82% with respect to the control mix. This is a value close to that reported by Algin and Ozen [73] for the addition of 0.10% basalt fibers in self-compacting concrete (SCC); they obtained a decrease of 13.0%. According to the RCPT results, the commercial grout CG-1 shows a high chloride ion penetrability. It was the mixture with the highest charge passed, and its permeability was high enough that before the 6 h that the test lasted, it exceeded the electrical charge measurable by the device, and the test was stopped 3 h after it started. The commercial grout CG-2 presents a permeability similar to the control grout, but it is noted that grouts with BF additions have a lower permeability than the commercial grout CG-2.

In post-tensioned and adhered construction, the corrosion protection provided by the grout becomes ineffective if the grout has high permeabilities, which allows moisture and chlorides to penetrate the steel tendon. Early cracking due to plastic shrinkage is often related to reduced durability of concrete structures [22], because these cracks produce pore interconnectivity and consequently increase permeability. This increase in permeable pores generates access routes for the permeation of water and chlorides, which would produce a risk of corrosion in the prestressing steel [22]. According to the results obtained, the addition of BF is effective to prevent plastic cracking, since a decrease in the permeability to chloride ions was found, which could be due to the decrease in interconnected pores.

## 4. Conclusions

Based on the methodology carried out and the results obtained under a rigorous measurement system that showed the implications of the use of basalt microfiber (BF) on the properties of the grouts for use in prestressed construction systems, the following conclusions were drawn:An inversely proportional relationship was found between the amount of BF and the flowability in the flow table. The mixture with the highest amount of BF obtained the lowest fluidity, which corresponds to grout BF-0.10 (0.10% BF) with a flow of 130.5%.The bulk density of the fresh grout mix had a decreasing trend with increasing BF volume content, the main reason being that BF makes entrapped air during mixing more difficult to remove.The addition of BF in percentages of volume from 0.03 to 0.10 reduces the permeability to air; this shows that the BF can control the early cracks that allow the interconnectivity of pores.Compared to the control grout, BF grouts slightly increase compressive strength at early ages. In advanced ages, this improvement is no longer noticeable, and the compressive strength is almost equal to that of control.Mixes with BF in the amounts of 0.03–0.10% by volume reduce drying shrinkage more than a shrinkage reducing additive alone can.A proportional trend with the BF content was found to decrease the rapid permeability to chloride. For this reason, the BF-0.10 mixture was the one that obtained the best results. This benefit of BF is that it prevents cracking in the plastic stage, and consequently decreases the number of interconnected capillary pores.In all the tests, the performances of the laboratory grouts were superior to those of the commercial grouts, especially in the permeability tests.The use of microreinforced grouts with BF is recommended in prestressed concrete pipes because, in quantities of up to 0.10%, sufficient fluidity is maintained to design a grout. In addition, BF does not decrease compressive strength and reduces possible microcracks and permeability.BF grouts benefit sustainability by building more durable prestressed structures. In addition, BF is an abundant and environmentally friendly material that can save production, maintenance and energy costs.

As a work in progress that gives continuity to this article, there is the adhesion test of the grout with BF to the prestressing steel tendon, creep tests and the implications of the fiber in the rheology of the paste.

## Figures and Tables

**Figure 1 materials-16-02842-f001:**
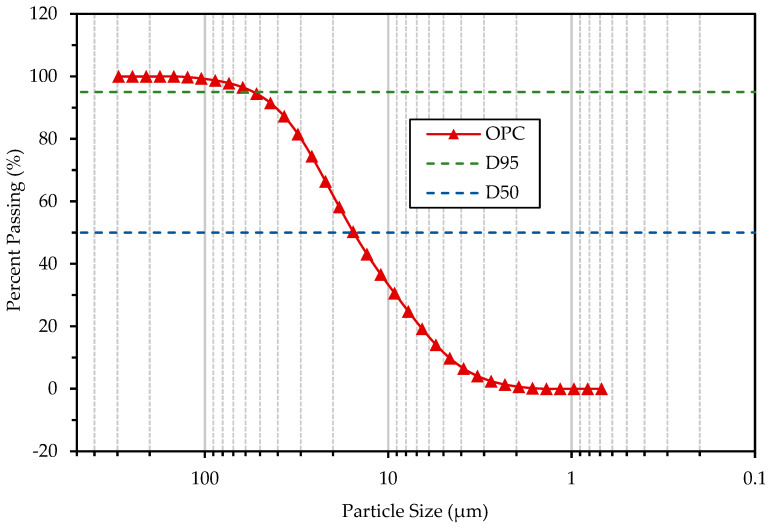
Particle size distribution of Ordinary Portland Cement (OPC). Obtained in laboratory.

**Figure 2 materials-16-02842-f002:**
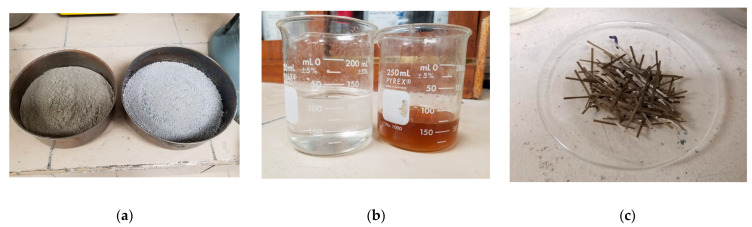
Materials, listed from left to right: (**a**) Ordinary Portland cement (OPC) and Fine Aggregate (FA); (**b**) additives SRA and SPA; and (**c**) basalt microfibers (BF).

**Figure 3 materials-16-02842-f003:**
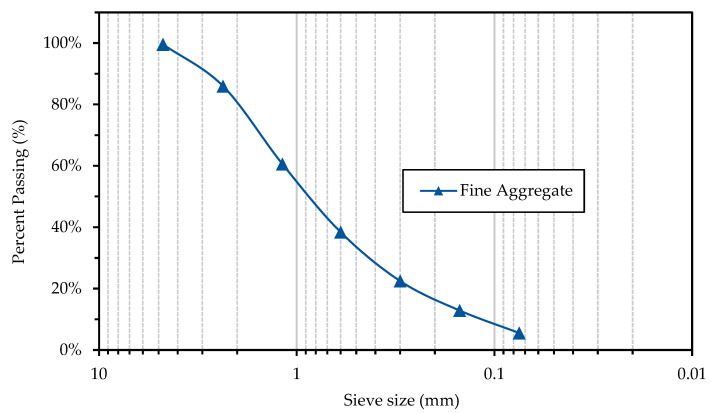
Sieve Analysis of Fine Aggregate (FA). Obtained in laboratory.

**Figure 4 materials-16-02842-f004:**
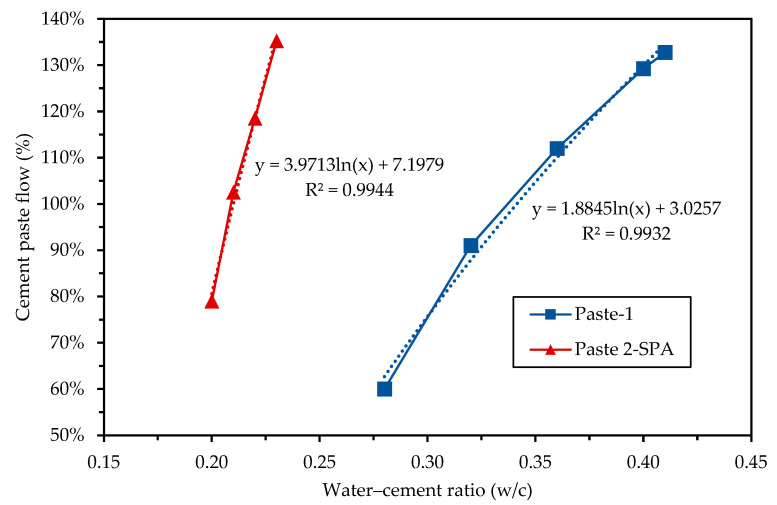
Influence of the water–cement ratio (w/c) and the superplasticizer additive (SPA) on the flow of cement pastes.

**Figure 5 materials-16-02842-f005:**
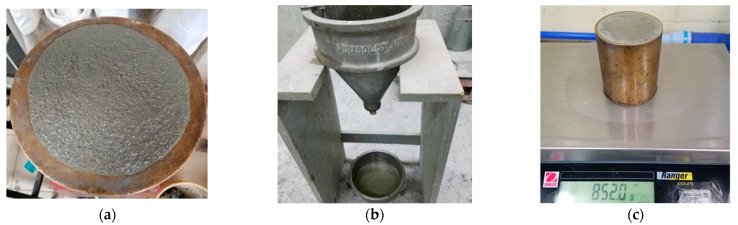
Tests in fresh state: (**a**) flow table ASTM C1437 (**b**) flow cone ASTM C939; and (**c**) unit mass and air content ASTM C185.

**Figure 6 materials-16-02842-f006:**
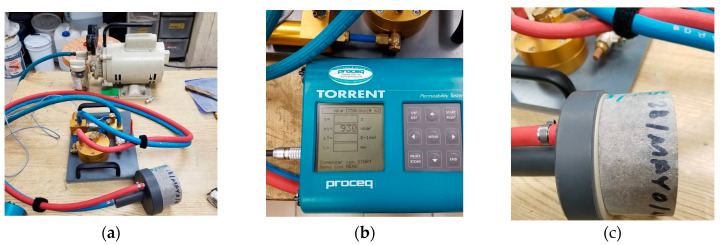
Air permeability tester: (**a**) vacuum pump and sensor (**b**) logger; and (**c**) chamber.

**Figure 7 materials-16-02842-f007:**
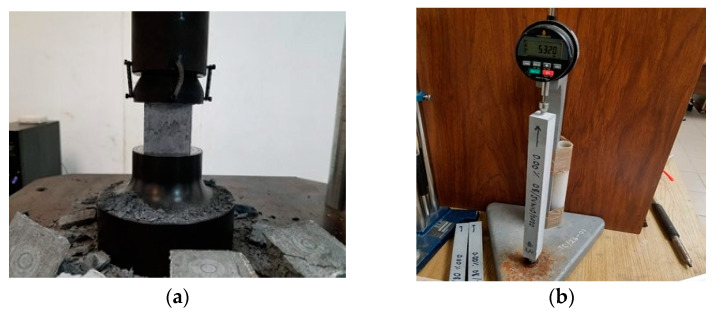
Mechanical resistance and volumetric changes: (**a**) compressive strength and (**b**) drying shrinkage.

**Figure 8 materials-16-02842-f008:**
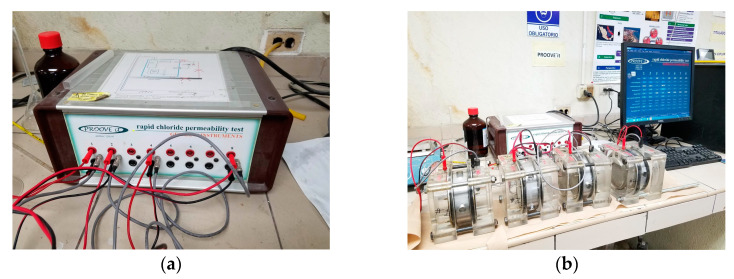
Durability indexes: (**a**) rapid chloride ion permeability test apparatus and (**b**) reservoirs and grout specimen.

**Figure 9 materials-16-02842-f009:**
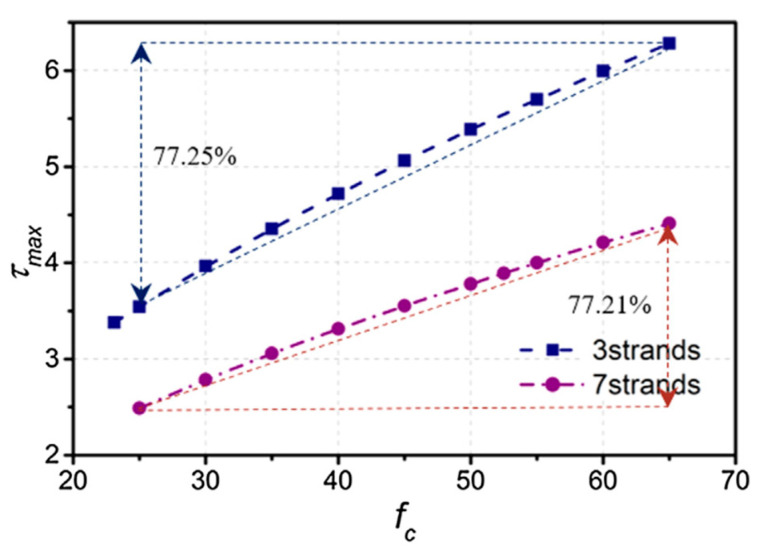
Effect of grout strength on the maximum bond stress [79].

**Figure 10 materials-16-02842-f010:**
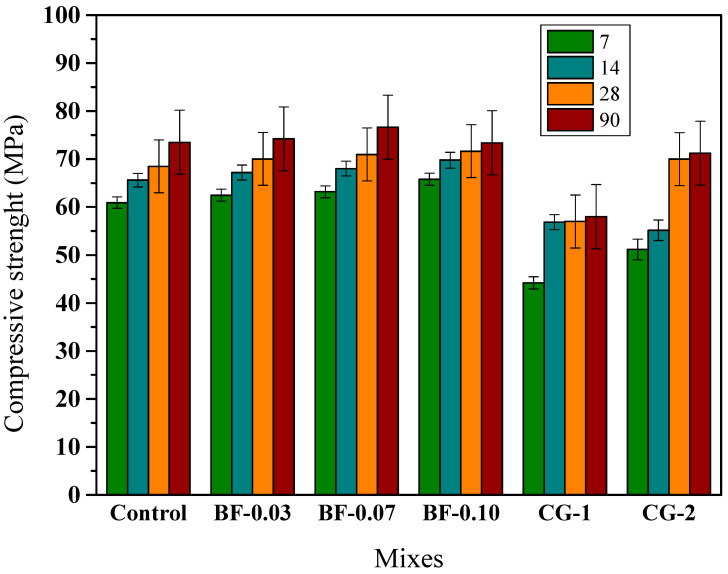
Compressive strength development up to 90 days of curing for grouts: control, BF-added and CG.

**Figure 11 materials-16-02842-f011:**
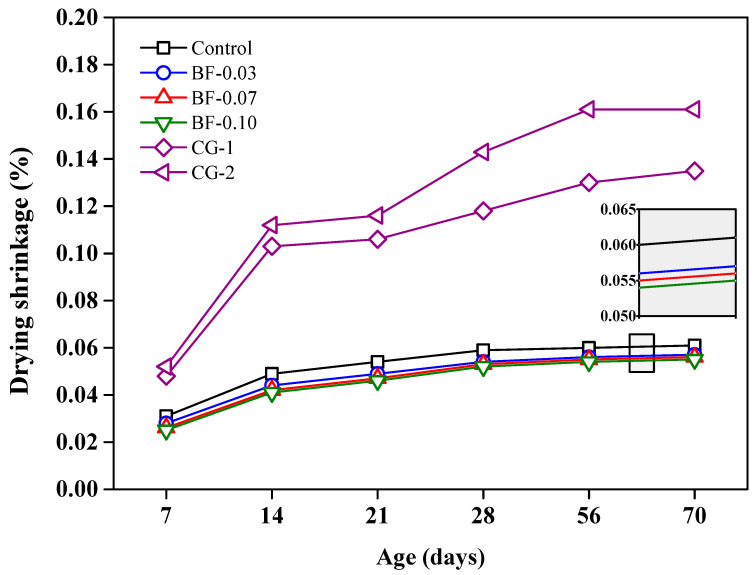
Dry shrinkage up to 70 days of curing for grouts: control, BF-added and CG.

**Figure 12 materials-16-02842-f012:**
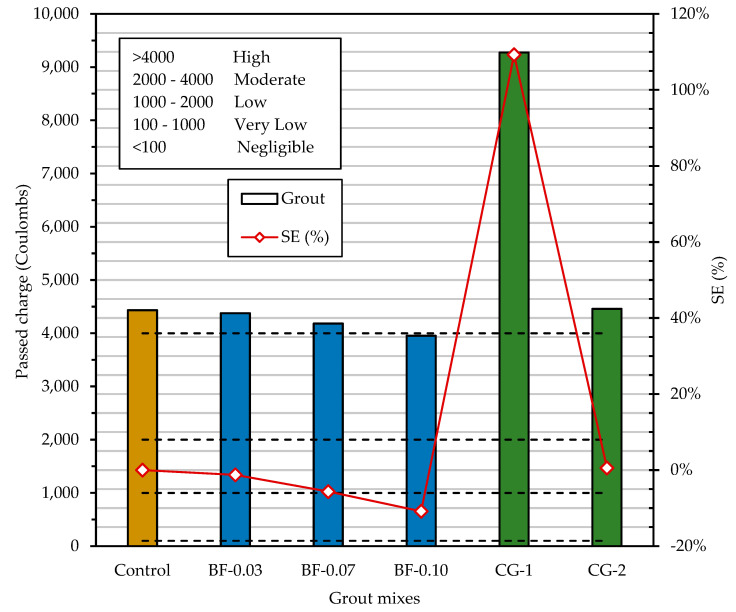
Rapid permeability to chloride ions in grouts, and the categories according to the passed charge in Coulombs. SE (%) = [(permeability of grout–permeability of control grout)/permeability of control grout] 100%.

**Table 1 materials-16-02842-t001:** Chemical composition and physical properties of OPC and CG. Obtained in laboratory.

Chemical Composition	OPC (%)	GC1 (%)	GC2 (%)
CaO	66.77	61.85	55.44
SiO_2_	20.16	26.38	29.30
Al_2_O_3_	1.72	1.31	2.01
Fe_2_O_3_	4.85	4.52	5.48
MgO	0.37	0.27	0.53
SO_3_	3.63	3.19	4.84
K_2_O	1.12	1.10	1.20
TiO_2_	0.36	0.30	0.40
Others	1.02	1.08	0.80
Relative density (g/cm^3^)	3.05	2.73	2.74

**Table 2 materials-16-02842-t002:** Physical properties of fine aggregate (FA).

Properties	Amount
Fineness modulus	2.80
Water absorption (%)	1.42
Density-Dd (g/cm^3^) ^a^	2.59
Density-Dssd (g/cm^3^) ^b^	2.63

^a^ Dd—absolute density in dry condition. ^b^ Dssd—absolute density in saturated and surface-dry conditions.

**Table 3 materials-16-02842-t003:** Physical and mechanical properties of BF. Obtained from manufacturer.

Diameter (µm)	Length (mm)	Elongation (%)	Density (g/cm³)	Elastic Modulus (GPa)	Tensile Strength (MPa)	Water Absorption (%)
13–15	21	3.1–3.2	2.70	93–110	2800–4800	<0.5

**Table 4 materials-16-02842-t004:** Mixing ratios of laboratory and commercial grouts.

Mixes	OPC (kg/m³)	FA (kg/m³)	Water (kg/m³)	SPA (%) ^a^	SRA (%) ^a^	BF (%) ^b^
Control	603	1413	241	0.61	2.00	--
BF-0.03	603	1413	241	0.61	2.00	0.03
BF-0.07	603	1412	240	0.61	2.00	0.07
BF-0.10	603	1411	240	0.61	2.00	0.10
CG-1	--	1833 ^c^	291	--	--	--
CG-2	--	1745 ^c^	346	--	--	--

^a^ %—percentage of solid content of additive with respect to the mass of cement. ^b^ %—percentage of basalt microfibers with respect to the volume of the mix. ^c^—includes the weight of the cementitious materials, aggregates and prepackaged additives of the commercial grouts.

**Table 5 materials-16-02842-t005:** Unit mass, air content and fluidity of the fresh grout mix. Average of 3 mixes.

Mixture	Density of Mixture (g/cm^3^)	Air Content (%)	Flow
Control	2.275	0.64%	>max allowed in the flow table
BF-0.03	2.246	1.83%	>max allowed in the flow table
BF-0.07	2.247	1.91%	>max allowed in the flow table
BF-0.10	2.231	2.53%	130.5% (ASTM C1107-grouts)
CG-1	2.129	3.50%	127% (ASTM C1437-mortars)
CG-2	2.088	1.86%	30 s (ASTM C939-liquid)

**Table 6 materials-16-02842-t006:** Quality of concrete according to the equipment manufacturer.

Concrete Quality	Index	KT (10^−16^ m^2^)
Very bad	5	>10
Bad	4	1.0–10
Normal	3	0.1–1.0
Good	2	0.01–0.1
Very good	1	<0.01

**Table 7 materials-16-02842-t007:** Results of the air permeability.

Mixture	KT (10^−16^ m^2^)	Grout Quality
Control	0.002	Very good
BF-0.03	0.001	Very good
BF-0.07	0.001	Very good
BF-0.10	0.001	Very good
CG-1	1.367	Bad
CG-2	0.163	Normal

## Data Availability

Not applicable.

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
