# Peer review of "Portland Cement-Based Grouts Enhanced with Basalt Fibers for Post-Tensioned Concrete Duct Filling"

_materials, 2023, doi:10.3390/ma16072842_

Round 1

Reviewer 1 Report

The author studied the Portland Cement-Based Grouts Enhanced with Basalt Fibers for Post-Tensioned Concrete Duct Filling. The research has engineering value. However, there are some problems in the analysis of the experimental details, which need to be improved, and the writing of the paper needs to be improved. Reviewers suggest reconsideration after major revision.

Abstract: In the beginning part of abstract, the significance of the research needs to be introduced.

Line 18-20, The specific advantages of this method compared with other methods are explained with detailed experimental data. For example, how much is the permeability reduced, and how is the pore connectivity?

In the introduction part, it is necessary to introduce the research work of relevant researchers on capability, such as https://doi.org/10.1016/j.cemconres.2020.106073.

Line 45-46, The two sentences before and after are illogical. It is necessary to add why the change of pore structure has a significant impact on cement-based materials. Related research can be added. For example, researchers have found that after adding materials, the pore structure and mechanical properties are improved. These two kinds of literature can be referred (https://doi.org/10.1016/j.jobe.2022.104880, https://doi.org/10.1007/s43452-022-00559-6).

Regarding basalt fiber reinforced concrete, it is necessary to introduce previous research work and highlight the advantages of this fiber.

Line73-76, The introduction of the research work is not detailed; please supplement the research content you have done.

Line80, OPC or PC, please check it.

Add the physical properties of cement, including specific surface area, standard strength, etc.

For SPA,What is the water reduction rate? Manufacturer information?

For Air permeability tester,Manufacturer information?

Line270-271,why evaluating the compressive strength is crucial to correlate a successful bond of grout and pre- 270 stressing steel.

For fig.5 and 6, Figures correspond to analysis, please modify.

The first letter of the coordinate axis title on the graph needs to be capitalized; please modify the full manuscript.

For Fig.6, The data is not clear; please modify it.

The conclusion is too long; please simplify the conclusion; 3-4 points are enough, and finally, look forward to the application prospect and value of this research.

Reviewer 2 Report

Reviewer Comments.

Portland Cement-Based Grouts Enhanced with Basalt Fibers for  Post-Tensioned Concrete Duct Filling

Post-tensioned concrete structures require a high-quality grouting material to fill the ducts and ensure proper transfer of tension forces. Portland cement-based grouts are commonly used for this purpose due to their strength and durability. However, these grouts may develop shrinkage cracks over time, which can compromise the long-term performance of the post-tensioned concrete structure. To address this issue, basalt fibers can be added to the grout to enhance its properties and reduce the likelihood of shrinkage cracks. The manuscript under review is related to work carried out on addition basalt fibers.

The significance of the work is high, as, thanks to the developed testing protocol and lot of experimental data, which was available and obtained. In this work, an original set of data is provided, that clearly relates with the title provided. The quality of the data is good, the work is of great scientific quality. Most part of the writing and the figures are also clear, however partly it is required to be improved. The article overall is written good, and authors describe the content of the article upto a great extent. In short, the work is considered as particularly good and linked to domain of this journal. However, some minor questions and comments follow for the clarification and improvement of the manuscript.

1-      Figure captions are too brief, author should emphasize and explain more about figures.

2-      What are the units of density in table 5?

3-      Basalt fibers and other materials pictures images are missing.

4-      Flowability of grout is important property and there is no result is concluded in conclusion section about the effect on flowabilty of grout due to addition of basalt fibers.

5-      The flow of grouts through the flow cone for all the grout is not given in the fresh properties of the grout. Revist this section and show the impact of blast fiber on flowability through the flow cone method.

6-      Authors need to revisit the compressive strength section of results and discussions where they needed to explain further about the comparison of compressive strength after the addition of all amounts of the basalt fibers in the mixes.

7-      In the compression section the literature review is giving about the optimum ratio of basalt fibers is 0.5%. then why others grouts contains only 0.1% of the basalt fibers.

8-      Results regarding adhesion properties of developed grout can also be further elaborated in the article.

I am hopeful that incorporation of these comments will further add certain value to the manuscript.

Thanks.

Regards.

 Dr. Muhammad Irfan-ul-Hassan

Reviewer 3 Report

The authors have presented an interesting paper, but it needs to be corrected before it can be published in the journal.

The introduction should be expanded to include a paragraph stating the objective and the novelty of this work compared to other research cited, as there are currently many studies using Basalt Fibers.

Indicate whether the results in Table 1 were carried out by the authors or extracted from the commercial brand. In the second case, indicate the source from which they were obtained.

Indicate the brand name of the SPA. 

Line 111 and others: "g/cm3" correct.

The results in Table 3, How were they obtained, indicate the standards used or whether they were taken from a manufacturer.

In Table 4, how was the water/cement ratio fixed? Has a standard been followed or has it been experimentally set to obtain a plastic consistency? Also, change "sand" to "aggregates".

Add the granulometric curve of the aggregates.

Table 5, add the units in Density of mixture

How many samples of each type were tested?

Why is shrinkage only measured up to 70 days, discuss the results presented in figure 6 based on other studies of fibre reinforced materials.

Include an introduction in the conclusions that discusses the implications of this work, as well as, a final paragraph showing limitations and future lines of work.

The bibliography is not in the journal format.

Round 2

Reviewer 1 Report

The author has seriously responded to all the questions raised by the reviewers. The overall answer is satisfactory, and the quality of the article has been dramatically improved. However, the citation of some references is inappropriate and needs to be revised. receive it after minor revision.

Line86-87, Too many references, some references are outdated, it is recommended to delete some references and replace some references,such as https://doi.org/10.1016/j.conbuildmat.2022.126921

Line96-97, Too many references, some references are outdated, it is recommended to delete some references and replace some references,such as https://doi.org/10.1016/j.conbuildmat.2022.130173

Author Response

We thank the reviewer for his comments and suggestions that help and contribute to the improvement of the manuscript.

Reviewer 3 Report

All the changes has been done

Author Response

(The authors gave the same response as above.)
